# Mapping Evidence on Management of Cervical Cancer in Sub-Saharan Africa: Scoping Review

**DOI:** 10.3390/ijerph19159207

**Published:** 2022-07-28

**Authors:** Petmore Zibako, Mbuzeleni Hlongwa, Nomsa Tsikai, Sarah Manyame, Themba G. Ginindza

**Affiliations:** 1Discipline of Public Health, School of Nursing and Public Health, University of KwaZulu-Natal, Durban 4000, South Africa; hlongwa.mbu@gmail.com (M.H.); ginindza@ukzn.ac.za (T.G.G.); 2Burden of Disease Research Unit, South African Medical Research Council, Cape Town 7505, South Africa; 3College of Health Sciences, University of Zimbabwe, MT Pleasant, Harare P.O. Box MP167, Zimbabwe; nomsatsikai@gmail.com (N.T.); smanyame79@gmail.com (S.M.); 4Cancer & Infectious Diseases Epidemiology Research Unit (CIDERU), College of Health Sciences, University of KwaZulu-Natal, Durban 4000, South Africa

**Keywords:** cervical cancer management, screening, diagnosis, chemotherapy, radiotherapy, human papillomavirus vaccine

## Abstract

Cervical cancer (CC) is the most common viral infection of the reproductive tract and in Sub-Saharan Africa (SSA), its morbidity and mortality rates are high. The aim of this review was to map evidence on CC management in SSA. The scoping review was conducted in accordance with Arksey and O’Malley’s scoping review framework. The review included studies on different aspects of CC management. The review was also done following the steps and guidelines outlined in the PRISMA-Extension for Scoping Reviews (PRISMA-ScR) checklist. The following databases were searched: PubMed, EBSCOhost, Scopus and Cochrane Database of Systematic Review. A total of 1121 studies were retrieved and 49 which were eligible for data extraction were included in the review. The studies were classifiable in 5 groups: 14 (28.57%) were on barriers to CC screening, 10 (20.41%) on factors associated with late-stage presentation at diagnosis, 11 (22.45%) on status of radiotherapy, 4 (8.20%) on status of chemotherapy and 10 (20.41%) on factors associated with high HPV coverage. High HPV vaccine coverage can be achieved using the class school-based strategy with opt-out consent form process. Barriers to CC screening uptake included lack of knowledge and awareness and unavailability of screening services. The reasons for late-stage presentation at diagnosis were unavailability of screening services, delaying whilst using complementary and alternative medicines and poor referral systems. The challenges in chemotherapy included unavailability and affordability, low survival rates, treatment interruption due to stock-outs as well as late presentation. Major challenges on radiotherapy were unavailability of radiotherapy, treatment interruption due to financial constraints, and machine breakdown and low quality of life. A gap in understanding the status of CC management in SSA has been revealed by the study implying that, without full knowledge of the extent of CC management, the challenges and opportunities, it will be difficult to reduce infection, improve treatment and palliative care. Research projects assessing knowledge, attitude and practice of those in immediate care of girls at vaccination age, situational analysis with health professionals and views of patients themselves is important to guide CC management practice.

## 1. Introduction

Cancer of the cervix is the most common cancer affecting women in Sub-Saharan Africa (SSA) [1]. Cervical cancer (CC) is caused by persistent infection with Human papillomavirus (HPV), whose genome proteins E6 and E7 inactivate tumor suppression genes pRb and p53 leading to cervical intraepithelial neoplasia (CIN) which will then develop into invasive CC [2]. It takes approximately 10 to 20 years from CIN to invasive CC [3]. Every year, about 500,000 women are diagnosed with CC worldwide and approximately 311,000 die from the disease [4]; 85% of the deaths occur in low-income countries (LICs) [5]. Incidence of CC is high with an average of 20.1% in SSA compared with 15.8% world-wide; CC related deaths at 13.8% for SSA and 8.2% worldwide [6]. Cervical cancer is insidious in its onset and slow in its progression but poses serious health outcomes like maternal and child mortality and morbidity. The pathway from symptom and signs recognition, care seeking, diagnosis and treatment is a process composed of events and processes that are influenced by different factors like demographics, health care system and disease factors.

Human papillomavirus infection is the most prevalent viral disease of the female reproductive tract [7]. Traditionally, CC prevention was composed of screening and early diagnosis but now, in high-income countries (HICs), HPV vaccine has become an integral part of CC prevention. The discovery of the HPV vaccine availed an opportunity to decrease morbidity and mortality associated with CC. Almost 99% of CC cases are caused by HPV, with HPV 16 and 18 causing seventy percent of the cases [7]. In Africa, there is limited access to screening and treatment [8] hence it is important to vaccinate girls against HPV as protection against CC. In SSA, preadolescent girls are not routinely included in the Expanded Program on Immunization (EPI) hence strategies for the delivery of the vaccine need to be explored. As of June 2018, only eight countries in SSA had a national HPV immunization program [9]. Rwanda reduced a two-decade gap in vaccination between HICs and LICs to five years [10]. In SSA, approximately 18.6 million school-aged girls are not in school [11] hence strategies to reach out to these girls are needed.

Early detection of CC is important because of the relationship between stage at diagnosis and survival rate. Women in developed countries have a two hundred and eight percent greater chance of being treated successfully for CC compared with women in Low- and Middle-Income Countries (LMICs) because of late presentation at diagnosis [12]. A five-year survival rate of 91% can be achieved if the patient is diagnosed with localized CC and the survival rate falls to 57% for regional and distant disease stages [13] hence the need to find out why women present with late-stage CC at diagnosis. Women who have HIV infection develop CC 10 years earlier than HIV-negative women; incidence of CC is about 900 per 100,000 in women with AIDS compared with 10 per 100,000 in the general population [14], hence the need to screen them yearly. Highly effective anti-retroviral therapy prolongs survival among those who are HIV-positive but does not prevent them from developing CC [15]. It is always important to keep checking how HIV positive women are coping with chemotherapy and radiotherapy, and to find any new evidence pertaining to CC treatment of such women.

In SSA, there is lack of timely cancer registries and incomplete risk factors for profiling [4]. Poor quality data curtail reliable population-based estimates for mortality rates, incidence rates and effectiveness of interventions. Of the twenty LMICs globally with the highest incidence of CC, sixteen are African countries [16]. The overall estimated standardized incidence rate is 35 per 100,000 women, and 23 per 100,000 women die from CC [17]. CC is a disease of the poor, showing evidence of inequities of access to health care [18]. CC mortality can be reduced by screening but ensuring high coverage of screening services is a serious challenge in LMICs, so there is a need to find out barriers to screening uptake. Coverage of CC screening is lowest in LMICs, estimated to be 19% [17]. Screening all women in a target age group every three years can prevent 91% of CC cases [19]. Screening is very crucial because CC takes 10–20 years for pre-cancer dysplasia to develop into invasive cervical cancer [14]. It is important to try and break barriers to screening after identifying the barriers.

CC patients in SSA present at an advanced stage which automatically precludes surgery as a treatment option in most patients [20]. Concomitant chemotherapy and radiotherapy with cisplatin as the radio-sensitizing drug is the standard of treatment for CC [21]. The efficacy of drugs for chemotherapy depends on delivering a full dose of treatment cycles for the required number of times continuously [22]. In SSA, women mostly present with advanced CC which needs standard curative treatment, which includes external beam radiotherapy, brachytherapy and with or without chemotherapy [14]. It is very crucial to keep checking the status of chemotherapy and radiotherapy services so as to maximize treatment outcomes based on available current treatment evidence.

Like chemotherapy, success of treatment using radiotherapy needs the completion of the cycles of the appropriate dose religiously. There is a potential for converting some patients from palliative to curative with radiotherapy and brachytherapy treatments, and the challenges facing such treatment options in SSA are worth solving [23]. Radiotherapy, like any other form of treatment, should improve the quality of life of CC patients [24]. Survival rate is another good measure of effectiveness of treatment [25]. Death rate per 100,000 women in SSA is twelve times higher than it is in Western Europe (25.3% vs. 2%) [26]. CC is a symbol of global health disparity [25] but can be reduced by finding the best method of treatment and prevention as well as finding out what causes these women to seek help late.

To improve care, it is important to understand the clinical epidemiology of CC and patient management pathways. The review aimed to map evidence on CC management in SSA. The review question was: what evidence is there on CC management in SSA? The objectives of the review were: To explore the factors that are associated with high coverage of HPV vaccine in SSA; to determine the barriers to CC screening uptake in SSA; to investigate the causes of late-stage presentation by CC patients at diagnosis in SSA; to establish the status of chemotherapy and radiotherapy services in CC management in SSA. The proposed hypothesis for this review was: CC management needs improvement in SSA.

## 2. Materials and Methods

This scoping review was conducted in accordance with Arksey and O’Malley’s scoping review framework [27] and the following steps were undertaken: (a) identifying the research question; (b) identifying relevant studies; (c) selecting relevant studies; (d) charting the data and (e) collating, summarizing and reporting the results. This review also adhered to the steps and guidelines outlined in the PRISMA-Extension for Scoping Reviews (PRISMA-ScR) checklist [28]. Protocol for this review was published a priori [29].

### 2.1. Identifying Relevant Studies

Based on the review question, a search strategy (Appendix A) was developed by identifying the key concepts using the PICO (Problem/Intervention/Comparison/Outcome) approach [30] and the search strategy was further developed using controlled vocabulary such as MeSH (Medical Subject Headings) terms. Articles published on CC management were reviewed for each of the following topics: CC prevention, detection (screening and diagnosis) and treatment. African country names and truncated terms such as ‘east Africa’ were also used to ensure that articles indexed using African country-specific names or regional terms were retrieved. The OR operator was used to combine synonyms, and the operator AND to filter the results which contain all the required terms. The Peer Review of Electronic Search Strategies (PRESS) checklist was used for the search strategy. The databases which were searched included: PubMed, EBSCOhost, Scopus and Cochrane Database of Systematic Review.

### 2.2. Study Selection

Two independent reviewers conducted abstracts and full article screening. The literature included published peer-reviewed journal articles with evidence of empirical design utilizing either qualitative, quantitative or mixed method research approach addressing the research questions. The screening procedure was guided by Higgins and Deeks’ framework [31]. All articles identified to be potentially eligible for inclusion in this review were obtained in full text. These articles were then exported to reference management software, Endnote (version 20, Stanford, CT, USA). Duplicates were removed before further screenings (abstract and full article) were conducted.

The inclusion criterion was guided by the following principles to determine articles relevant for this review: Studies presenting evidence on CC or factors associated with high HPV vaccine coverage, factors associated with late-stage presentation at diagnosis, barriers to screening uptake, and status of chemotherapy and radiotherapy services; studies presenting evidence conducted in SSA. No limits were applied for the publication dates of included studies and all study designs were considered. Studies that did not focus on humans, as well as those written in languages other than English were excluded. Non-empirical material like book chapters, opinion papers, commentaries and editorials were not included.

### 2.3. Data Extraction and Charting

A data extraction instrument (Appendix A) was developed to confirm study characteristics as well as relevance. Data was extracted by the principal investigator. The data extraction form included the following elements: author(s), year of publication, title of study, country, study aim, study design, study population, sample size, key findings that relate to the review question, study limitations and conclusions from the authors. Data were entered into Microsoft Excel and qualitative data was uploaded in NVivo version 10 [32], a computer-assisted qualitative data analysis software.

### 2.4. Data Analysis

A narrative synthesis [33] was used, with data synthesized and interpreted using sifting, charting and sorting based on themes and key issues. Citation tracking was done using Reference Manager Software in Endnote version X7. Unstructured texts extracted from the articles were exported into NVivo for qualitative analysis. Thematic and descriptive analyses were used in summarizing and identifying patterns across studies. Textual data summary was tabulated from qualitative, mixed methods and quantitative studies. The directed content analysis method was used on abstracted data to identify patterns or themes that characterize factors that affect CC management. Meta-analysis and tabulation of the findings was done through Review Manager (RevMan) [34] using the random effects model, because there was variability in study designs, sampling and effect measures explored. Meta-analysis results were presented using tabular format and interval forest plot. The z-test was used to test for the overall effect at 5% level of significance.

### 2.5. Quality Control and Assessment

Studies that were published between the research and report writing were obtained by subscribing to updates to databases using the search domains used during literature search. Data were extracted by the principal investigator and accuracy was checked by a second reviewer. Studies with uncertainties about their inclusion were discussed with a third reviewer. Data were collected from May 2020 to October 2020. The quality of evidence was assessed based on guidance in National Institute for Health and Care Excellence single technology appraisal Specification for Manufacturer/Sponsor Submission of Evidence adapted from the Centre for Reviews and Dissemination’s guidance for undertaking reviews in health care [35].

Mixed Method Quality Appraisal Tool (MMAT) was used for quality assurance [36]. The checklist was used disregarding inapplicable criteria. All eligible studies were assessed for quality of the study and quality of reporting with aspects like: sampling frame, stating hypothesis, defined target population, defined study population, stated study setting, dates in which the study was conducted, eligibility criteria, selection into the study, justification of number of participants, stated number of participants at the beginning of the study, methods of data collection, reliability/repeatability measurement, methods of follow up, were participants at each stage specified, were the reasons for loss to follow-up quantified, was missing data accounted for in the analysis, was the impact of bias estimated quantitatively, and what were used to assess the quality of included studies. The studies were rated as good quality with comments on each study. Dissemination of the results need is by publications in a journal and presentation at health conferences. The quality of studies’ assessment is found on Appendix A.

### 2.6. Screening of Studies 

The PRISMA (Preferred Reporting Items for Systematic Review and Meta-Analysis) flow chart was used to display screening decisions and results [37] (Figure 1).

Total studies identified 1121 (959 PubMed, 50 Scopus, 100 EBSCOhost, and 12 Cochrane), and effectively 49 remained for the scoping review.

Of the 49 studies included:14 were on barriers to CC screening,10 on factors associated with late-stage presentation at diagnosis,11 on status of radiotherapy,4 on status of chemotherapy and10 on factors associated with high HPV coverage.

The electronic search strategy identified 1121 references (Figure 1), which were screened for titles. A total of 201 duplicates were removed, leaving 920 articles which were screened for abstracts. A total of 865 articles were removed at the abstract screening stage because they formed part of the exclusion criteria. The researchers further screened 55 full-text articles and excluded 6 for the following reasons: 4 did not have adequate sample size, 1 study was not done in SSA, and 1 study where the full text was not in English. Therefore, 49 articles met our inclusion criteria and were included in the quality assessment stage.

A total of 49 studies were included during the review analysis (see Appendix A for a summary of the studies).

## 3. Results

Twenty-nine percent of the articles were covering barriers associated with screening uptake, while 20% [10] were covering factors associated with high HPV vaccine coverage, 8% [8] on chemotherapy, 23% [11] on radiotherapy, and 20% [10] were covering factors associated with late-stage presentation of CC at diagnosis. The median year of publications included was 2015 ranging from 2007 to 2020. At least 41% [20] of the study designs used in these articles were cross-sectional studies, 21% [10] were qualitative studies (focus groups and in-depth interviews), 16% were retrospective cohort studies, 11% [5] were case studies or series, 9% [4] were prospective cohort studies, and only a single randomized controlled trial study was included in the analysis. These studies were carried out in SSA which included the following countries: Kenya [9], South Africa [5], Malawi [5], Ethiopia [5], Uganda [4], Tanzania [3], Ghana [4], Zimbabwe [4], Rwanda [2], Nigeria [3], Mozambique [1], Botswana [2], Senegal [1], and Cameroon [1]. Sample sizes ranged from 15 to 98, 792 participants. Nine studies reported on incidence of CC, ranging from 22.0 to 75.9 per 100,000 women with a median value of 35.1/100,000 as shown in Figure 2.

Cervical cancer mortality rates were reported by 7 studies whose values ranged from 17.5 to 49.8 per 100,000 women with a median of 36.0 per 100,000, as displayed in Figure 3.

Survival rates ranged from 13% to 29% with a median of 17.7% from 7 studies. The statistics are displayed in Figure 4.

### 3.1. Factors Associated with High HPV Vaccine Coverage

Adolescents, guardians, key informants and important stakeholders were among the study population for the 10 articles that were included in the analysis of factors associated with high HPV vaccine coverage. Among those articles that reported the HPV vaccine coverage levels, the pooled estimate of the HPV vaccine coverage level was estimated to be 86% and a 95% confidence interval ranging from 81% to 90%, as shown in Figure 5.

Of these articles, 50% reported that a class school-based program is among the significant factors associated with high HPV vaccine coverage levels. Seventy percent of the articles cited the use of various forms of media to be associated with high HPV vaccine coverage levels; 30% cited the use of various forms of capacity building programs to be associated with high HPV vaccine coverage levels. On the other hand, 50% cited that high knowledge, positive attitudes and perception levels are associated with high HPV vaccine coverage levels.

Outreach strategies like school-based reduce institutional challenges for those willing to take up the vaccine [10]. Age-based vaccination was inferior to class-based vaccination which located more girls for vaccination and achieved higher coverage [7,49]. The experience of reaching slum-dwellers and pastoralists for implementing HPV vaccine program is feasible in hard-to-reach communities, in addition to the school-based strategy, where the vaccine should be offered at multiple centers using a campaign method [50].

Involvement of the local media in promoting the vaccine, scientific information on efficacy as well as adverse events was also important [51]. Importing vaccine through MOH, WHO or UNICEF to minimize administrative costs and import duties was necessary for high coverage [51]. There was an opt-out consent approach whereby parents indicate to teachers that they do not want their daughter to be vaccinated, and challenging parental refusal if a daughter wished to receive the vaccine through community leaders like chiefs. Running the HPV vaccine program alongside well-known programs like a de-worming program was needed for high coverage [52]. The rest of the factors that were associated with high HPV vaccine coverage are presented in Table 1.

Acceptability of the HPV vaccine does not translate to high HPV uptake unless operational problems are addressed [53]. Uses of school-based strategies reduce operational costs for all stakeholders [52]. School-based strategy is the best but depends on high school attendance [46].

### 3.2. Factors Associated with Late-Stage Cancer Presentation at Diagnosis

CC patients were the only study population for the 10 articles that were included in the analysis of factors associated with late-stage CC presentation at diagnosis. Among those articles that reported the prevalence of late-stage CC cancer, the pooled estimate of the prevalence of late-stage CC cancer was estimated to be 71% and a 95% confidence interval ranging from 60% to 82% (see Figure 6).

Of these articles, 50% cited that low knowledge, negative attitudes and perception were associated with presentation with late-stage CC, 40% cited that poor financial resources are associated with presentation with late-stage CC, 20% cited that distance and accessibility of health care facilities were associated with presentation with late-stage CC, 50% cited that low education level was associated with presentation of late-stage cancer, 20% cited that older age was associated with presentation of late-stage cancer, 20% cited that negative involvement of the marital partner was associated with presentation at late- stage cancer, and 30% cited that lack of CC screening was associated with presentation of late-stage cancer.

Most patients presented with late-stage squamous cell type and the majority of the patients belonged to the low socio-economic status class [55]. Factors such as unemployment, residing in rural areas, and lack of medical insurance were mentioned to be also associated with presentation of late-stage cancer [56]. Factors associated with late-stage CC presentation at diagnosis are shown in Table 2.

### 3.3. Barriers to CC Screening Uptake

CC patients, health care workers, HIV patients, important stakeholders and general women were among the study population for the 14 articles that were included in the analysis of barriers for utilization of CC screening. Among those articles that reported the prevalence of CC screening, the pooled estimate of the prevalence of CC screening was estimated to be 12% and a 95% confidence interval ranging from 7% to 17% as shown in Figure 7.

Of these articles, 86% [12] reported that poor knowledge, negative attitudes and perceptions were among the significant barriers of CC screening services uptake. Seventy-one percent cited poor financial constraints as a barrier, 64% [10] cited issues to do with long distance or inaccessibility to the health care facility as a barrier, 43% [6] cited issues to do with traditional beliefs or myths as a barrier, 50% [7] cited issues to do with lack of infrastructure as a barrier, and 61% [9] cited issues to do with inadequate screening staff or use of male staff in the provision of screening services as a barrier.

Factors such as religious followings, negative involvement by a partner, low level of education, lack of screening facilities, and residing in rural areas were also mentioned to be associated with underutilization of CC screening services [9]. Institutional barriers included: over-burdened health care facilities with lack of equipment, understaffed and negative attitude towards CC by health professions and lack of disposable speculums [54]. A comprehensive list of barriers to CC screening is shown in Table 3.

### 3.4. Status of Chemotherapy in SSA

CC patients and health professionals were among the study population for the 4 articles that were included in the analysis stage of chemotherapy status. Factors such as poor financial resources, availability and accessibility of chemotherapy, side effects like anemia, renal dysfunction and stage of presentation with cancer, were mentioned to be associated with initiation or the outcomes of chemotherapy [21]. Some of the outcomes measured were survival, quality of life, eligibility and feasibility of chemotherapy treatment [38]. Additionally, there was the theme of stock-outs affecting treatment outcomes and, consequently, survival rate as well as the need to have a cancer registry with correct epidemiological data so as to improve procurement of chemotherapy [2]. The use of cheap generics in SSA also affected availability as manufacturers are not motivated to manufacture such drugs [59]. There was also a theme of religion, beliefs and culture affecting chemotherapy uptake where some believed that CC was a death sentence hence no need to go for chemotherapy because it was not effective or the treatment can kill the patient and leave the family in poverty because of its expensiveness [43]. Most patients came for treatment when only palliative care was the only treatment feasible, after they had wasted time on traditional healers and religious healing methods [60]. The rest of the aspects of chemotherapy are shown in Table 4.

### 3.5. Status of Radiotherapy in SSA

CC patients and health professions were among the study population for the 11 studies that were included in the analysis of radiotherapy status. Factors such as poor affordability, availability of infrastructure, side effects and late-stage at presentation were mentioned to be associated with initiation or the outcomes of radiotherapy [39]. Some of the outcomes measured were survival, quality of life, eligibility and feasibility of radiotherapy treatment [44]. CC treatment with radiotherapy was constrained, with the major challenge being the scarcity of radiotherapy services due to breakdown of the machine, which was not available at all in some countries; lack of brachytherapy as well as the fact that the machines were outdated and they needed new ones which include modern technology which reduces side effects by not exposing healthy cells to radiation [61].

In some countries, the radiotherapy machine was available but was not operational [40]. The financial burden of CC treatment and lack of insurance was one of the important predictors to an abandonment of treatment [62]. Long distances to access diagnostic and treatment services, lack of decentralized diagnostic and treatment facilities have been established as factors hindering access to CC treatment [63].

Patients consider physical functioning more important when appraising health status compared to QOL, where mental health is emphasized [64]. Evidence showed that all the domains of sexual functioning, sexual activity and enjoyment as well as vaginal functioning declined after treatment [14]. The radiotherapy machine had frequent breakdowns and there was no back-up for the machine as it is a very expensive piece of equipment [8]. There was a lot of treatment interruption which resulted in poor survival rates and high mortality rates [8]. Late presentation and negative cultural beliefs also affected radiotherapy outcomes [4]. Some of the aspects that describe the status of radiotherapy in SSA are listed in Table 5.

## 4. Discussion

The HPV vaccine is an acceptable form of CC prevention and high coverage can be achieved using the school-based strategy, with out-of-school girls being reached using local health centers [41,48,53,65]. Reasons for late presentation included unavailability of screening, delaying seeking health care whilst going to traditional healers and faith healers, fear of CC diagnosis, health professional misdiagnosis, lack of knowledge about CC and negative cultural beliefs [12,55,57]. Barriers to screening uptake were lack of knowledge about CC, unavailability of screening services, fear of positive results, misconception about screening procedures and negative cultural beliefs [42,54]. The status of chemotherapy was bad, characterized by unavailability, affordability and accessibility challenges [22]. The situation was worsened by most patients presenting with late-stage disease, side effects and lack of investigating services due to financial reasons [15,20]. The major outcomes were poor survival rates and high mortality rates [21]. Radiotherapy status was bad, with the major aspect being the unavailability of radiotherapy services due to high demand, machine breakdown and the outdated aspects of the machines exposing patients to a lot of side effects [25,26,43].

The school-based strategy works well in countries like South Africa where primary school level is compulsory and universal [47]. The grade to be chosen should be based on the fact that the girls to be vaccinated are not yet sexually active and have reached an appropriate age to understand sexual education. Community involvement such as including chiefs and religious leaders and social mobilization, micro planning, health promotion and health informatics were major contributors to high coverage [9]. For the good of the girls, consent process might need to be waived, especially where religion and culture are the limiting factors. Opt-out vaccine consent process produced higher coverage compared to opt-in models [47]. Teachers can be empowered to be vaccine champions in disseminating information about HPV vaccine as well as CC in their community. HPV vaccination program needs a well-established vaccine delivery mechanism with adequate transportation, cold chain, human resources and capacity to monitor the whole process.

More still needs to be done in SSA as far as the HPV vaccine rollout; by December 2017, only 3 countries, South Africa, Uganda and Rwanda, had transitioned from the pilot to national program [11]. Vaccine hesitance was mainly due to the fact that the vaccine was relatively new, that it will cause the girls to be promiscuous and that it was considered inappropriate to target young girls to prevent sexually transmittable diseases [46]. A positive attitude towards the HPV vaccine was a strength which contributed to high coverage. Mother–daughter approach is not an efficient way to deliver vaccine since maternal CC screening is more time consuming than HPV vaccine and in one of the studies, 3000 women were screened versus 2000 girls that were vaccinated [8]. Vaccination was seen as a culturally acceptable form of CC prevention as some women found the practice of CC screening embarrassing, too intimate and uncomfortable [60].

Absence of a nationally organized screening program (opportunistic screening) and lack of money for CC treatment caused women to not seek health care early until symptoms become worse. The average prevalence of late-stage presentation was 71%. The Model of Pathways to Treatment (MPT) explains well how the delays occurred through: appraisal, help-seeking, diagnostic and pre-treatment [39]. Public awareness campaigns for the women and continuous professional development (CPD) for the health care professionals can be used to deal with these delay intervals.

The core strategies to prevent late-stage presentation by CC patients include population-based CC screening and prompt treatment of pre-invasive cervical lesions [39]. Poorly differentiated histology is an intrinsic tumor characteristic which cannot be easily modified hence the need to intensify screening and reduce both patient and institutional delay. Spending time trying to treat cancer using traditional medicines was cited as a factor contributing to late presentation by CC patients [44] hence the need to train traditional healers with basic training to alert them to CC symptoms so that they can quickly refer women with CC to health care centers. Community leaders must also be educated about CC for them to be able to give accurate information to assist in identification of CC which is often overlooked.

Health professionals were also a contributing factor to late-stage presentation as some health personnel were not aware of CC. It took 6 to 12 months for referrals to take place [61]. Women cannot solely be responsible for late presentation because lack of suspicion of CC by health care professionals and the lack of prioritization of CC management by health departments were major contributors to presentation of advanced stage at diagnosis [40]. Women’s health issues that are not related to maternity or family planning lacked priority. Information regarding the link between CC and sexual activity is important to allow women to make an informed decision about their sexual behavior.

Due to molecular interactions between HIV and HPV, CC is an HIV/AIDS defining disease [62,63], hence the need to integrate routine HIV care with CC screening as HIV- positive women are recommended to be screened for CC yearly [64]. Routine reminders and appointments for screening for HIV-positive CC patients are needed instead of relying on the patients’ initiatives. Cell phones can be used for such reminders. Standard doses for chemo radiotherapy can be considered as the standard of care for appropriately selected HIV-positive women with CC [21]. The role of CC survivors in advocacy and mobilization for CC treatment and screening cannot be overemphasized since knowing someone with CC was associated with knowledge about CC and high levels of screening [41]. Lack of knowledge about CC was a major barrier to CC screening, hence the need to increase health education about CC which is hoped to translate to an increase in screening uptake but knowledge does not always translate to practice as in some studies where many of the health professions were never screened themselves [57].

Awareness campaigns and education must be undertaken by health professionals since motivators for screening included the doctor’s recommendation, fear of death from cancer and affordability [42]. A new curriculum for health care professionals is needed which includes prevention and intervention for both non-communicable and communicable diseases. CC screening discussion should be mandatory between health workers and women whenever they seek any health care. The health education has to be user-friendly, as 8 million people in SSA were not able to benefit from existing written health promotion material due to illiteracy [58]. Educational campaigns must focus on increasing risk perceptions, improving attitudes and educating women to seek screening when they are free from signs and symptoms of CC, as it was found out that success of the screening program depends largely on CC knowledge and health-seeking behaviors [66].

Cost of screening was cited as a barrier to screening uptake so there is a need to subsidize screening costs or make it completely free since some studies included transport costs as a screening barrier [45]. Religious and cultural factors were a barrier to CC screening uptake as they affected health-seeking behaviors and practices like early marriages, and onset of sexual activities present the need to protect the girls by using the HPV vaccine. Socio-cultural beliefs resulted in women’s perception of low threat of CC. An individual’s perceptions on health issues can also hinder health behavior like CC screening [67], hence the need to educate women. CC awareness needs to consider varying religious and cultural beliefs in order to implement an effective CC screening program.

The unavailability of screening services was a major barrier so there is a need to offer screening services from Monday to Friday so that clients are not turned away because the services are offered on certain days. Governments in SSA need to honor their commitment to the Abuja declaration which provides for 15% of their national budget to health [38]. Lack of epidemiological data, cancer service policies, human resources, financial resources and political will were barriers which can only be addressed by the government. The incidence rates ranged from 22/100,000 to 76/100,000 women, and mortality rate ranged from 18/100,000 to 50/100,000; such figures can only be reduced with strong political will and mobilization of resources to build capacity for screening and treatment [68].

Chemo radiotherapy using cisplatin as the radio-sensitizing agent was the standard of care for CC treatment and all effort should be made to avoid tenofovir because of overlapping, neurological, renal toxicities and hematologic with cisplatin [21]. The human life loss due to stock-outs of chemotherapy drugs that are used to treat CC is quite significant since efficacy of these regimens depends on taking a full dose per schedule for a required frequency of treatment cycles; this includes administering fewer cycles, delays in cycles or absence of any of the drugs which will reduce the curative potential to zero [22]. Interruption of treatment can cause the patient to go out of remission and in some cases, remission will not be achieved on reinstitution of the drug and the ultimate result is an unnecessary loss of life. A low proportion of CC patients benefit from chemotherapy due to late presentation in developing countries, hence the need for a national screening program as opposed to opportunistic screening [15]. This late presentation at diagnosis also precludes the use of surgery as a treatment option for CC [20].

In Africa, Ethiopia has the second largest gap, after Nigeria, between the availability of and demand for radiotherapy machines; judging from the WHO recommendations, there are 73 radiotherapy units missing in Ethiopia which had just one machine for the whole country [26].

Breakdown of radiotherapy machines was a major problem in CC treatment since this causes treatment interruptions [68]. There is a need for a clear policy to deal with treatment interruptions. Cost of chemo radiotherapy was the most difficult challenge faced by CC patients which can be sold by the governments in SSA, subsidizing CC treatment since most patients cannot afford it. Despite increased toxicity, chemotherapy and radiotherapy are widely accepted as a major treatment for CC [67].

Deterioration of quality of life occurs because of the diagnosis of late-stage CC and due to treatment, hence the need for management services that ensure coping with CC for both patients and home-based caregivers [44]. Counseling must be integrated in management of CC patients and relatives to enhance coping at all stages of CC care. Expensiveness of chemotherapy resulted in it not being regularly included in chemo radiotherapy; this lack of therapeutic options and early detection activities resulted in low probabilities of survival like a range of 5-year survival of 2.9% to 22% in SSA compared to 68% in the USA [25]. The exclusion criteria for a patient to undergo chemo radiotherapy included anemia, hydronephrosis and impaired renal function [45]. The other factors that contributed to low survival rate in SSA included factors like the 28 countries which did not have a radiotherapy facility, and where they did have, the radiotherapy machine was not adequate; 30% of radiotherapy machines were Cobalt60 units instead of intensity modulated using linear accelerators which are the standard of care as they reduce side effects due to unnecessary irradiation of the surrounding tissues [26]. Brachytherapy was available in 20 of 52 African countries [26].

The key factors that cut across all themes are as follows:limited or absence of necessary infrastructure and financial resources,behavioral issues of patients—religion, societal view shaping the behavior,lack of knowledge and skills—on the side of patients and health professionals, respectively,poor planning and governance on the side of governments in these countries.

### 4.1. Strength and Limitations of the Study

The study benefited from a systematic approach to identifying studies and reviewing them with no country, population, age or date limitation. The Arksey and O’Malley’s framework, as one of the best scoping review frameworks, was used. The scoping review results were presented in line with PRISMA-ScR recommendations, which advance transparent and exhaustive reporting. The review included all types of designs ensuring inclusion of all possible data on CC management within SSA. We acknowledge that exclusion of papers that were published in other languages other than English is a limiting factor, and some insightful papers have not yet been published. This study is also not spared from publication bias; however, effort was made to ensure that wide spectrums of databases were used for the search. The study search terminology covered the whole spectrum of CC management; however, some terminology may exist that were not used in this study, resulting in possible omission. The inclusion of MeSH terms has the strength to overcome that possible omission bias.

### 4.2. Implication for Research

Future studies need to investigate the gaps identified in this study, exploring them at a country level considering the experience and views of patients and health professionals. In addition, the need to include those in the care of girls who are at the vaccination stage [9,10,11,12,13,14] is essential to help improve vaccination uptake. As noted in this study, beliefs, knowledge gaps and attitude have the possibility of hindering CC management; investigating deeply at each individual country level is therefore critical as cultural differences and practices do differ markedly from country to country. Some studies looked at more than one country, ignoring the possible heterogeneity in population characteristics across countries. There were limited studies on therapeutic impact of chemo radiotherapy, hence the need to do some prospective cohort studies on CC management.

### 4.3. Implication for Practice

From the policy perspective, provision of free vaccine is needed with such services available at the primary health facility level. All health professionals need to have ample knowledge about non-communicable diseases like CC so as to avoid delays in the referral system and thus avoid late-stage presentation. Quantification of chemotherapy can be conducted using data from the National Cancer Registry and a resource-sensitive treatment strategy to help reduce stock-outs and produce an effective and efficient procurement system. Radiotherapy centers should strive to acquire at least two radiotherapy machines to minimize treatment interruption. Setting fiscal allocation targets towards capacitating and resourcing the health system in line with the needs for CC management is vital in SSA. Waging a war against CC, as has been the case with HIV/AIDS pandemic, is what is now needed—there is a need for the political will, and women in leadership are better placed to help with lobbying and resource mobilization. A national screening program and the provision of radiotherapy services are major priorities needed in SSA. Subsidization is required as most patients did not have insurance coverage. Current best practice for CC should include brachytherapy.

## 5. Conclusions

The authors were able to assess the extent of CC management in SSA and it was observed that CC management is lagging in all categories. Despite that CC is among the top cause of death, worldwide, in developing countries and in SSA, with seemingly less knowledge about its management, a dearth of studies exist. The review points to the need for dedicated funding towards CC management to ensure health facilities are well resourced and capacitated (equipment, human capital, drug stock). A gap was identified in the literature, on the knowledge, attitude and practice (KAP) of those in care of girls at vaccination stage (9–14 years), the experience and views of patients and the assessment of the state of CC from a health professional’s point of view. Current literature often bundles the patients’ and non-patients’ views, yet understanding patient factors, behavioral characteristics and health care factors that are associated with late-stage presentation at diagnosis are important for effective public health intervention, prompt health-seeking behavior and ultimately will improve survival. Provision of quality palliative care is a missing component of CC management in SSA, and the suggested future research projects will help to assess the challenges and opportunities to improve the outcome. In addition, the dearth of studies can be remedied by further studies in chemo radiotherapy therapeutics of CC.

## Figures and Tables

**Figure 1 ijerph-19-09207-f001:**
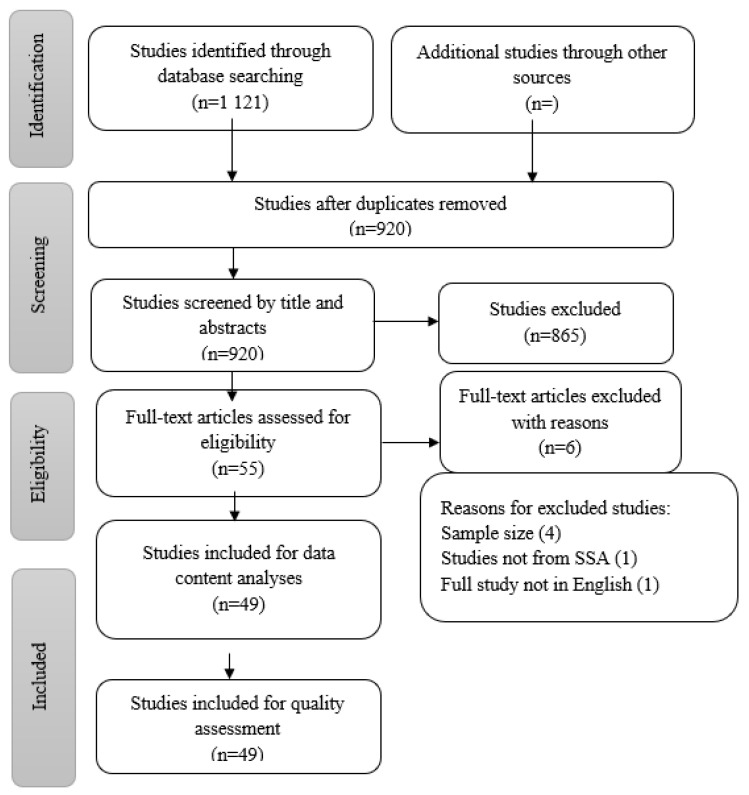
PRISMA flow diagram of the study selection process.

**Figure 2 ijerph-19-09207-f002:**
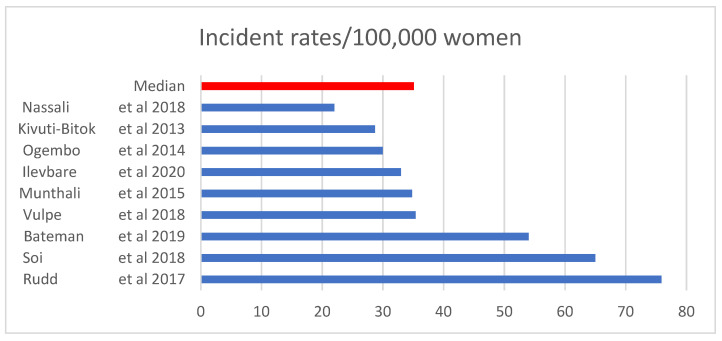
Cervical cancer incidence. Sources: [8,11,38,39,40,41,42].

**Figure 3 ijerph-19-09207-f003:**
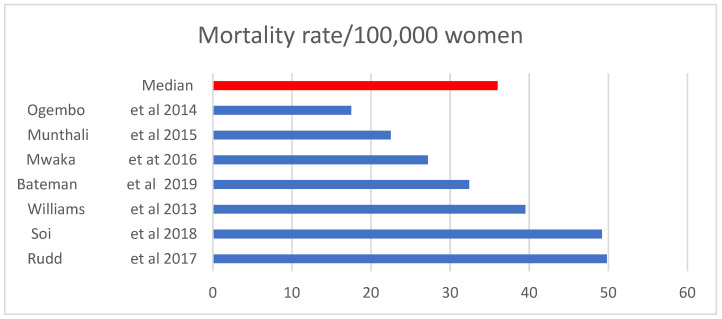
Cervical cancer mortality rate. Sources: [8,11,40,42,43,44,45].

**Figure 4 ijerph-19-09207-f004:**
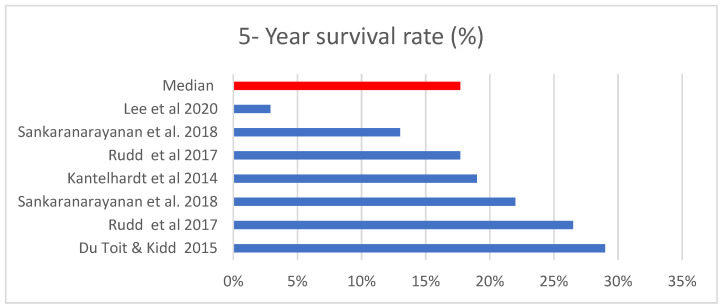
Cervical cancer survival rate. Source: [13,25,40,46].

**Figure 5 ijerph-19-09207-f005:**
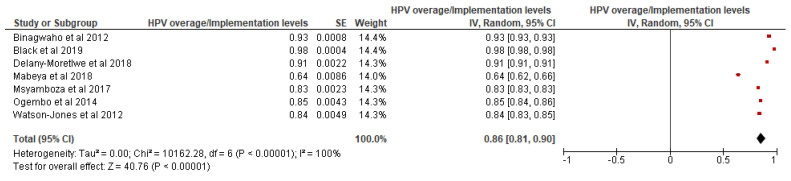
HPV vaccine Coverage: Pilot studies. Source: [7,8,9,10,37,47,48].

**Figure 6 ijerph-19-09207-f006:**
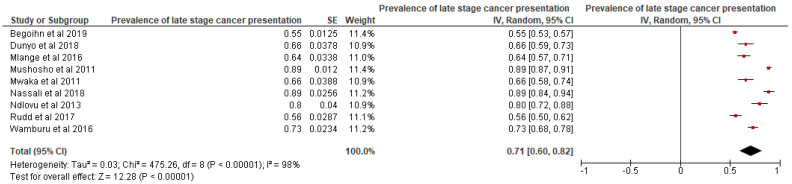
Prevalence of late-stage cancer presentation at diagnosis. Source: [12,40,41,44,50,54].

**Figure 7 ijerph-19-09207-f007:**
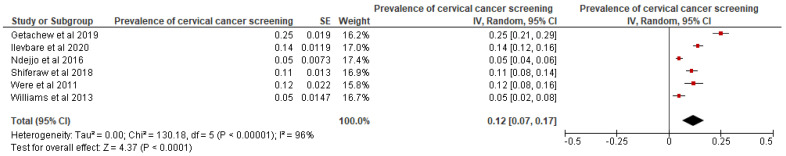
CC screening uptake Rates. Source: [1,17,38,45,57,58].

**Table 1 ijerph-19-09207-t001:** Factors associated with high HPV vaccine coverage.

Comprehensive and careful planning, Strong commitment by government, Early community sensitization and outreach/social mobilization, Involving both ministries of health and education for school-based HPV vaccination. Collaboration between private and public institutions in terms of strong ownership. Communication of role expectations of all stakeholders and streamlining consent processes. Education of parents and guardians on CC for both men and women, since both are involved in vaccination decision-making. Empowering teachers to be vaccine champions and in disseminating information on HPV vaccine and CC. Community involvement in identifying girls out of school and those absent from school during vaccination. Speeches by health professionals, government, clergy and other community leaders to advise about HPV vaccine. Multiple phases to vaccination and providing extra vaccination opportunities for adolescent girls who missed schedules. Financial support from international partners such as GAVI. Use of mixed method approaches to vaccine delivery. Mother–daughter approach in hard-to-reach girls. Extending CC screening to mothers to motivate them to take their daughters to vaccination centers. Thorough explanation of vaccine benefits, safety, and risks of CC to leaders and pastors in their villages and churches. Support and peer tracking of girls by leaders and pastors. Use of an electronic database to enable recalling girls for follow-up and for monitoring and evaluation. Health workers’ beliefs in the importance of HPV vaccines. Good infrastructure like roads and the existence of bridges as well as an economically stable status. Organizational incentives and rewards. Involvement of the local media in promoting the vaccine, scientific information on efficacy as well as adverse events. Permitting stakeholders to ask questions and provision of honest and evidence-based answers. Importing vaccine through MOH, WHO or UNICEF to minimize administrative costs and import duties. Use of mobile clinics among girls in hard-to-reach remote areas. Hospital-based to reduce costs of the program for opportunistic vaccination. Outreach by health staff to encourage and inform about HPV vaccination. Verbal as well as written information on HPV vaccine through school, community meetings, print and radio, drama/plays. Opt-out consent approach whereby parents indicate to teachers that they do not want their daughter to be vaccinated.

**Table 2 ijerph-19-09207-t002:** Factors Associated with late-stage cancer presentation at diagnosis.

Long patient intervals Lack of previous screening Ignorant of symptoms of CC Non-functioning or inadequate screening Low socio-economic status class Unmarried status Intrinsic tumor characteristics Lack of pathological capacity Lack of suspicion of CC by health care professionals and the lack of prioritization of CC management by Health Departments Stigma, misinformation and lack of financial resources/lack of screening opportunities High age, rural residence, low level of education/financial and logistics, poorly differentiated tumors, no prior screening Lack of awareness and misinterpretation of the seriousness of symptoms Lack of medical insurance/early onset of sexual activity among low educated women Lack of specificity of CC symptoms and inadequate facilities for diagnosis Health care professionals’ misinterpretations of CC symptoms and subsequent diagnosis of non-CC conditions Illiteracy of patients Poor community and health care worker awareness of CC; overbooked clinics Treatment from traditional healers, herbalists as well as religious healers Lack of knowledge Social isolation Fear of exposure during screening with association of CC and promiscuity Sex of the clinician and age of the health profession involved in screening Some women did not report main symptoms at consultation Poor referral system of the health care system Cultural beliefs that abnormal vaginal bleeding was caused by witchcraft and that the body is cleansing itself Lack of specialized health practitioners Absence of frequent gynecological examinations Lack of awareness on the importance of regular gynecological examinations Opportunistic screening with lack of quality control systems and poor coverage Patient delay, practitioner delay and system delay Women assumed symptoms resulted from the continuation of menses, irregular menses and genital infections Fear associated with pain from a Pap smear Lack of cytology laboratory, arrangements to communicate results to screened women and facility for confirming the diagnosis

**Table 3 ijerph-19-09207-t003:** Barriers to CC screening uptake.

Living in rural areas Low service provider recommendations to go for screening services Expensive screening methods and transport cost to screening facilities Fear and stigma surrounding CC, lack of information and access to screening services Socio-cultural beliefs that diagnosis leads to death Fear that screening could lead to discovery of CC Lack of knowledge and inadequate infrastructure Lack of training and motivation as CC screening was treated as tusk shifting Cost of screening Negative health personnel attitude, lack of privacy and misdiagnosis Socio-cultural belief about the etiology of CC; belief in traditional medicine Lack of funding at policy level Normative gender relations and need for approval of partner to undergo screening Lack of government subsidy on CC screening Lack of policies on the management of CC or poor implementation of the existing policies Unavailability and inaccessibility of screening facilities and traveling long distances to health centers Myths and misconceptions that a woman’s ovaries and uterus were being removed during screening Fear of pain associated with CC screening, fear of undressing to preserve privacy and low perceived CC risk Shortage of health professionals to routinely do CC Shyness as well as embarrassment to expose private parts to health professionals The belief that CC is caused by a promiscuous lifestyle and the belief that CC was a punishment from gods Preference for divine intervention instead of screening Fear of stress from an additional diagnosis Long waiting time; inadequate space Competing health priorities as well as low prioritization of CC screening Lack of information; public awareness The beliefs that CC is caused by the breach of social taboos Cost of screening, the pain of the procedure and being attended by male medical staff Poor attitude towards screening Limited training among health care providers Lack of health care insurance and poverty in general Health care systems are donor-funded and focus on specific diseases like TB, HIV, malaria and maternal health Not having required screening skills and equipment Large workload compromised quality of care given to patients seeking screening services Influence of husbands and in-laws Social inequalities in rural areas, poor road conditions, lack of public transport Poorly supervised, lack of basic equipment and stock-outs of basic medical supplies in health facilities; inadequate funding No standards and guidelines for CC screening

**Table 4 ijerph-19-09207-t004:** Chemotherapy status in SSA.

Majority of HIV-infected and non-infected women with CC can complete chemo radiotherapy with the same cisplatin dose Rampant chemotherapy stock-outs A small proportion of women with CC would benefit from chemotherapy because of late presentation at diagnosis Chemotherapy and analgesics were not affordable and were not available Lack of blood for transfusion The most common adverse event was decreased lymphocyte count Adverse events in treated participants included: diarrhoea, vomiting, chronic kidney disease, syncope, hypermagnesemia Standard doses for chemo radiotherapy can be considered as standard of care for selected HIV-positive women Concomitant chemo radiotherapy using cisplatin is the standard of care for the treatment of CC in SSA Tenofovir should be avoided because of potentially overlapping neurological, hematologic and renal toxicities with cisplatin Funding was inadequate to cover pharmaceuticals needed for CC treatment and CC diagnosis annually Lack of standardized treatment protocols limited the ability to predict prescribing patterns Patients experienced suboptimal therapy due to delays in therapy, missed doses, substitutions Shortages were related to weak infrastructure for the procurement and erratic availability of chemotherapy and stock- outs Alignment with WHO National Essential Medicine List for SSA was 34.1% Diversion from standard care due to drug stock-outs as well as differences in doctors’ prescribing preferences. Chemotherapy drugs’ pricing market is highly variable and not transparent Quantification of chemotherapy needed can be done using data from National Cancer Registry Most countries did not know where they have the correct epidemiological data Resource-sensitive treatment strategy helps reduce stock-outs as well as to produce efficient procurement systems The efficacy of chemotherapy regimens depends on delivering the full dose on schedule Treatment interruption causes patient to go out of remission Common exclusion criteria for chemotherapy were hydronephrosis and anemia HIV-positive patients were more likely to meet multiple exclusion criteria Concomitant chemo radiotherapy produces an overall survival advantage of 10–16% in the treatment of CC Failure to establish eligibility for chemo radiotherapy was due to economic, geographic, social and psychological factors HIV-positive women fared worse because of advanced stage at presentation and had low probability of completing treatment Some received no chemotherapy due to poor renal function Renal dysfunction was the common reason for not completing chemotherapy Chemotherapy component is the difficult aspect of chemo radiation for HIV-positive patients to complete Patients who failed to complete chemotherapy had lower CD4 counts than those who completed it Commonly used chemotherapy drugs were cisplatin and 5-fluorouracil The most prevalent histological type of CC was squamous cell carcinoma (SCC) (90%) Combined EBRT, brachytherapy and chemotherapy had significantly higher gastrointestinal acute toxicity than EBRT alone. No deaths occurred directly due to acute treatment toxicity Due to limited availability and finances, less than 10% of palliative patients received additional chemotherapy For palliation, chemotherapy included cisplatin, paclitaxel and/or bevacizumab as the standard of care Treating health facilities do not provide chemotherapy drugs and they refer patients to private pharmacies.

**Table 5 ijerph-19-09207-t005:** Radiotherapy status in SSA.

Treatment interruption Poor documentation of patients’ records (non-computerized) and no mechanism for patient follow-up Chemo radiation improved quality of life better than radiation only in certain situations Chemo radiation is the treatment option in situations where quality of life is the goal of treatment High-dose brachytherapy implementation is possible in developing countries with fixed geometry applicators Low capacity for external-beam radiation and brachytherapy with some countries not having such facilities Advanced stage at presentation is the main prognostic factor for low survival Radiation doses had higher survival rates compared to lower doses Later stages had lower survival rates compared to earlier stages which were higher Poverty, lack of education, lack of awareness, absence of screening programs caused late presentation at radiotherapy facilities Late presentation at diagnosis, sub-optimal treatment and diagnosis are major factors causing low survival rate of patients Socio-economic reasons and lack of radiation capacities caused low survival rates Travel and hygiene maintenance costs for CC patients CC patients who needed special financial assistance included stage IV, HIV-positive, widows, and those with minimal education CC treatment had negative effect on the QOL in all domains of lives of women with CC Health systems present barriers to access of CC treatment and general care Promising outcomes were seen in women with CC who were treated with chemo radiation therapy together with brachytherapy The radiotherapy facility serves the whole country as well as neighboring countries for some SSA countries Due to frequent breakdowns of the machine, patients were booked 4 months down the line Majority of patients did not access radiotherapy due to distance, cost and heavy booking at radiotherapy National screening program and the provision of radiotherapy services are major priorities needed in SSA Most patients experienced treatment interruption due to financial challenges, machine breakdown, side effects There is a need for a clear policy to deal with treatment interruptions Government should give subsidy for CC management Intravenous pyelogram (IVP), magnetic resonance imaging (MRI), cystoscopy were not performed due to limited funds Most radiotherapy did not include the application of brachytherapy Chemo radiation therapy improved quality of life better than radiation therapy only in certain situations Palliative radiotherapy was administered to almost half of the patients because of the lack of finances Change in FIGO stage between pathological diagnosis and the start of radiotherapy, during which time a number of patients died while waiting for treatment Some patients experienced recurrence after treatment (28%) Only 10% received optimal combined EBRT, brachytherapy and adjuvant chemotherapy, with the majority receiving only EBRT Patients who received combined EBRT, brachytherapy and adjuvant chemotherapy had better tumor control and survival CC cases were very advanced at presentation and treatment outcomes were poor Reasons for discontinuation were toxicities, economic background, and radiotherapy machine breakdown Discontinuation of planned radiotherapy reduced survival for all stages treated Side effects of radiotherapy included radiation proctitis, dermatitis, diarrhea, subcutaneous fibrosis and vaginal stricture For other associated diagnostic and routine laboratory tests, the patients would be referred to private facilities Most of the patients depended on relatives and church members for financial assistance, yet all with no formal employment Swelling of the feet as lymphoedema is a complication of pelvic radiotherapy which was experienced by most patients Late side effects are vaginal stenosis/shortening, proctitis, hematuria, subcutaneous fibrosis, vesicovaginal fistulas Reasons for interruptions included severe anemia or neutropenia, GI toxicity, machine breakdown

## Data Availability

Not applicable.

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
