# Peer review of "Mapping Evidence on Management of Cervical Cancer in Sub-Saharan Africa: Scoping Review"

_ijerph, 2022, doi:10.3390/ijerph19159207_

Round 1

Reviewer 1 Report

Thank you for offering me the opportunity to review this manuscript. The scoping review "Mapping Evidence on Management of Cervical Cancer in Sub-Saharan Africa: Scoping Review" does provide an important contribution to the literature on detection, prevention, and treatment of Cervical Cancer. Congratulations to the authors for the work done.

Please download the attached document to read my comments.

Author Response

Thank you for the comments; all comments have ben addressed as per attached 

Reviewer 2 Report

The manuscript entitled " Mapping evidence on management of cervical cancer in sub-saharan Africa: Scoping and review" is written well and concise.

The authors extensively documented the screening, identification approaches. More importantly the manuscript widely covers about vaccination, barriers to screening, chemotherapy and radiotherapy.

This kind of review article are much needed in the field of study to improve the cervical cancer treatment.                      

The manuscript has scientific merit but some points could be improved. I recommend this manuscript for publication after the following corrections and suggestions.

Throughout the manuscript there are several typos, the authors should clearly check the spell checks, space etc...

1. For eg: line number 75 and 85... there should be space "per 100,000".

2. Throughout the manuscript the tables should be numbered and I recommend to put the comprehensive planning in boxes.

3. Some of the sentences are starting with small letters, the authors should change accordingly.

Author Response

Dear Reviewer,

Thank you for the review and thorough feedback. All comments have been addressed as per attached file.

Reviewer 3 Report

Well written and comprehensive review.

It would be relevant if some of the supplemental data can be converted to figures and graphs to be included into the text, for example, inbcidence and mortality rates.

So also screening, coverage with vaccination, completion of chemotherapy and radiation can be represented in graphic form.

Author Response

Dear Reviewer,

Thank you for the feedback; all comments have been addressed as per attached.

Kind Regards
